# Differences between Treadmill and Cycle Ergometer Cardiopulmonary Exercise Testing Results in Triathletes and Their Association with Body Composition and Body Mass Index

**DOI:** 10.3390/ijerph19063557

**Published:** 2022-03-17

**Authors:** Szymon Price, Szczepan Wiecha, Igor Cieśliński, Daniel Śliż, Przemysław Seweryn Kasiak, Jacek Lach, Grzegorz Gruba, Tomasz Kowalski, Artur Mamcarz

**Affiliations:** 13rd Department of Internal Medicine and Cardiology, Medical University of Warsaw, 02-091 Warsaw, Poland; szymonprice@gmail.com (S.P.); jacek.lach@wum.edu.pl (J.L.); artur.mamcarz@wum.edu.pl (A.M.); 2Department of Physical Education and Health in Biala Podlaska, Faculty in Biala Podlaska, Jozef Pilsudski University of Physical Education in Warsaw, 21-500 Biala Podlaska, Poland; igor.cieslinski@awf.edu.pl; 3Public Health School Centrum Medyczne Kształcenia Podyplomowego (CMKP), 01-813 Warsaw, Poland; 4Students’ Scientific Group of Lifestyle Medicine, 3rd Department of Internal Medicine and Cardiology, Medical University of Warsaw, 02-091 Warsaw, Poland; przemyslaw.kasiak1@gmail.com (P.S.K.); gregorygpl@gmail.com (G.G.); 5Institute of Sport-National Research Institute, 01-982 Warsaw, Poland; tomekbielany@gmail.com

**Keywords:** triathlon training, heart rate, ventilation

## Abstract

Cardiopulmonary exercise testing (CPET) is the method of choice to assess aerobic fitness. Previous research was ambiguous as to whether treadmill (TE) and cycle ergometry (CE) results are transferrable or different between testing modalities in triathletes. The aim of this paper was to investigate the differences in HR and VO_2_ at maximum exertion between TE and CE, at anaerobic threshold (AT) and respiratory compensation point (RCP) and evaluate their association with body fat (BF), fat-free mass (FFM) and body mass index (BMI). In total, 143 adult (*n* = 18 female), Caucasian triathletes had both Tr and CE CPET performed. The male group was divided into <40 years (*n* = 80) and >40 years (*n* = 45). Females were aged between 18 and 46 years. Body composition was measured with bioelectrical impedance before tests. Differences were evaluated using paired *t*-tests, and associations were evaluated in males using multiple linear regression (MLR). Significant differences were found in VO_2_ and HR at maximum exertion, at AT and at RCP between CE and TE testing, in both males and females. VO_2AT_ was 38.8 (±4.6) mL/kg/min in TE vs. 32.8 (±5.4) in CE in males and 36.0 (±3.6) vs. 32.1 (±3.8) in females (*p* < 0.001). HR_AT_ was 149 (±10) bpm in TE vs. 136 (±11) in CE in males and 156 (±7) vs. 146 (±11) in females (*p* < 0.001). VO_2_max was 52 (±6) mL/kg/min vs. 49 (±7) in CE in males and 45.3 (±4.9) in Tr vs. 43.9 (±5.2) in females (*p* < 0.001). HRmax was 183 (±10) bpm in TE vs. 177 (±10) in CE in males and 183 (±9) vs. 179 (±10) in females (*p* < 0.001). MLR showed that BMI, BF and FFM are significantly associated with differences in HR and VO_2_ at maximum, AT and RCP in males aged >40. Both tests should be used independently to achieve optimal fitness assessments and further training planning.

## 1. Introduction

Cardiopulmonary exercise testing (CPET) is a dynamic, non-invasive method to assess the cardiopulmonary system at rest and during exercise [1]. It may be applied in medicine to evaluate the degree of cardiovascular function impairment and plan rehabilitation and in sports science to assess participants’ fitness [2]. Key variables measured in CPET include heart rate (HR), oxygen consumption (VO_2_), respiratory rate (RR), pulmonary ventilation (VE), oxygen pulse, respiratory exchange ratio (RER), ventilatory equivalents for oxygen (VE/VO_2_) and carbon dioxide (VE/VCO_2_) [3]. The most frequently compared variable is the maximum oxygen uptake (VO_2_max), which may be defined as the highest value reached, despite progressive increase of the load applied, with the development of a plateau in the VO_2_ [3]. Another important value is the VO_2_ at the anaerobic threshold (AT), corresponding to the threshold between moderate and high-intensity exercise, which is the point when the lack of sufficient oxygen supply to the exercising muscles necessitates glycolytic ATP production and the accumulation of lactic acid [4]. Thus, in exercise at an intensity below AT, lactate remains at resting levels, while in high-intensity exercise above AT, lactate rises until an elevated steady state is attained [4]. The respiratory compensation point (RCP) is identified as the second breakpoint in the ventilation response and is a measurable variable most closely related to the concept of critical power (CP), which in turn represents the point separating power outputs that can be sustained for a prolonged time from power outputs, which lead to a certain maximum after which exercise intolerance occurs [4,5]. CP is especially relevant in high-intensity training or intermittent highintensity training [5].

The most commonly used testing modalities are the cycle ergometer (CE) and treadmill (TE), with various protocols or self-paced [6,7]. These training modalities both have unique strengths and weaknesses. The TE activates more muscle groups, and VO_2_max is generally higher than in the cycle ergometer by 7–18%, varying between studies, although there exist many conflicting papers, and results are inconsistent [8]. The cycle ergometer allows better electrocardiographic (ECG) analysis due to fewer artefacts from upper body motion [6]. The relationship between testing modality and VO_2_ max is ambiguous. Triathletes with previous experience in cycling may obtain results on the cycle ergometer that are equal to or even higher than those obtained on the treadmill, while trained runners display higher results in treadmill testing [9,10,11]. Usually, for triathletes the testing modality is selected specifically to fit the discipline trained by the examinee, i.e., a treadmill for runners or a cycle ergometer for cyclists [12]. A unique challenge is posed by triathletes, who have no single mode of training, but rather devote a portion of their training to swimming, cycling and running [8]. The monitoring of training would therefore ideally be carried out with all of the specific tests for the most accurate results, but this would be highly impractical given that testing is time consuming, costly and must be repeated regularly [12]. It is therefore important to know whether there is a significant difference between treadmill and cycle ergometry results in triathletes. Few studies have been conducted to assess this, and existing studies often included small studied groups of fewer than twenty participants [8]. The results of previous studies are inconclusive, showing that VO_2_ max in triathletes may be equal [13,14,15,16,17], higher [18,19] or even lower [20] in treadmill testing compared with cycle ergometry. The anaerobic threshold (AT) and lactate threshold (LT) were also either reported as higher, in the treadmill test [21], or similar, in both tests [22]. Millet et al. point out that study methods were often unclear and the study group sizes were limited in many of the existing studies [8].

Maximal heart rate (HRmax) is either reported as similar [15,18,21,23], or slightly higher in treadmill testing compared with cycle ergometry [20,21,24]. It is also unclear whether this relationship is true for males and females alike, or only for males [8]. Another important parameter to consider is the HR corresponding to the AT, which is often used to prescribe submaximal exercise training loads [8]. This value has previously been generally reported as higher in treadmill tests compared with cycle ergometry [15,24,25,26]. However, some studies yielded no significant difference [13,27].The sex differences between triathletes in running and cycling are also unclear. Most studies did not evaluate these differences at all, and the ones that did yield no difference between cycling and running for both males and females [16,28].

It is also unclear how age affects the differences between testing modalities. Aerobic capacity decreases rapidly after the age of 40 years in males and is related to muscle mass [29,30,31]. Older age is also associated with a decreased exercise efficiency and an increase in the oxygen cost of exercise, which contribute to a decreased exercise capacity. These age-related changes may be reversed with exercise training, which improves efficiency to a greater degree in the elderly than in the young [32]. It has been proposed that the difference in VO_2_max between treadmill and cycle ergometry between runners and triathletes may be due to the higher muscle mass of triathletes, especially in the upper body, and not to running economy [8]. Body composition has been shown to impact triathlon performance. Fat mass and fat percentage are positively associated with race time (i.e., the race time is greater in participants with higher fat mass), while fat-free mass is negatively related to race time [33]. Another paper demonstrated that body fat is associated with race time in male Ironman triathletes but not in females [34].

To the best of our knowledge, no studies evaluated whether an association exists between body composition (BC) and differences in VO_2_max in different testing modalities in triathletes. Despite previous research on the topic, it remains unclear whether treadmill and cycle ergometry may be used interchangeably for the monitoring of training in triathletes, largely due to insufficient data on the differences in results obtained from both testing modalities [8,23,35].

The main aim of this study was to assess the difference in VO_2_ and HR at maximum exertion and at AT and RCP in cycle ergometry and treadmill testing in triathletes of various levels. A further aim was to evaluate whether an association exists between these parameters and BC or body mass index (BMI). Based on previous literature, we hypothesized that results are different in different testing modalities in triathletes and that they are influenced by body composition.

## 2. Materials and Methods

### 2.1. Participants Preliminary Inclusion Criteria

The study involving human participants was reviewed and approved by the Bioethical Committee of the Medical University of Warsaw. The patients/participants provided their written informed consent to participate in this study. All procedures were carried out in accordance with the Declaration of Helsinki. The data of participants was gathered from records of commercial CPET performed in the years 2013–2020. They were recruited via the internet and social media advertisements or via recommendations from trainers or other clients. The tests were carried out on the personal request of the participants as part of training optimization and diagnostics. The participants were triathletes who had participated in competitive events. Inclusion criteria for the study were: age over 18 years, triathlon training for at least three months, having a treadmill test and a cycle ergometer test performed within a maximum two months’ timeframe and meeting the maximum exertion criteria described below. Exclusion criteria were any chronic or acute medical conditions (including musculoskeletal system disorders such as new fractures and sprains, as well as addiction to nicotine, alcohol or other substances) or the ongoing intake of any medication. Identical study methods and procedures were used during the entire period from which data were gathered. Participants were informed via e-mail on how to prepare for the test. They were advised to avoid any exercise 2 h prior to the test, eat a light carbohydrate meal 2–3 h before the test and stay hydrated by drinking isotonic beverages. They were also instructed to avoid medicines, caffeine and cigarettes before the test.

### 2.2. Selected Subjects

From the database, we obtained 238 individual cases of people who carried out the study twice (cycling and running separately). After verifying the inclusion criteria, we obtained 143 cases included in further analysis. The average time interval between both tests was 2.44 ± 3.10 days in female and 5.29 ± 6.81 in male triathletes. The order of testing was random. For 86 cases, running protocol was the first test performed. Populational data were calculated as means with standard deviation (SD) and are presented in Table 1. Documented competition experience from the earliest competition to the day of the first test was an average of 94.1 ± 38.8 months; 95% CI from 74.2 to 113.9 in females and an average of 103.3 ± 42.8 months and 95% CI from 95.6 to 110.9 in male triathletes (Table 2). The population was also divided according to age into two groups, <40 years and >40 years (age was not included as an independent variable in these models).

### 2.3. Measures

Body mass (BM) and fat mass (FM) were measured with the use of a BC analyzer (Tanita, MC 718, Tokyo, Japan) before every test with the multifrequency 5 kHz/50 kHz/250 kHz electrical bioimpedance method. The BC tests were conducted directly prior to each CPET if the interval between tests was >48 h, and mean values from both tests were further analyzed. In cases where two CPET tests were carried out on the following days, only one BC analysis was performed prior to the first CPET. All measurements (BC and CPET) took place under similar conditions in the medical clinic Sportslab (www.sportslab.pl; accessed on 2 February 2022, Warsaw, Poland). The conditions were 40 m^2^ of indoor, air-conditioned space, altitude 100 m MSL, temperature 20–22 degrees Centigrade and 40–60% humidity.

### 2.4. CPET Equipment

Exercise tests were performed on a cycle ergometer Cyclus−2 (RBM elektronik-automation GmbH, Leipzig, Germany) and on a mechanical treadmill (h/p/Cosmos quasar, Germany), within one day–two months of one another. During all tests, cardiopulmonary indices were recorded using a Cosmed Quark CPET device (Rome, Italy), calibrated before each test according to the manufacturer’s instructions. HR was measured using the ANT+ chest strap, which is part of the Cosmed Quark CPET device (declared accuracy similar to ECG, ±1 bpm.). The Cosmed Quark CPET software has been updated regularly over the years (from PFT Suite to Omnia 10.0 E.). During the entire data collection period, three Cosmed Quark CPET gas analyzers were used (each replaced after three to four years of use). All the mechanical equipment used in CPET testing procedures was serviced and checked by the producer every year to keep their technical passports valid in accordance with local regulations for medical facilities. Each test was preceded by a 5 min adaptation (walking or pedaling with no resistance). To account for the different exercise capacity of the triathletes, the initial power (Watt) or speed (km/h) were determined based on an interview carried out before each individual test. The lowest power at which the participant subjectively felt resistance was selected as the initial power for cycle ergometer tests (60–150 W). The power was then increased by 20–30 W every 2 min. For treadmill tests, the start speed was an individually selected slow running pace, between 7 and 12 km/h based on the interviews, and 1% incline was applied. The speed was then increased by 1 km/h every 2 min. To assess the maximum level of aerobic fitness, participants were instructed to maintain the effort for as long as possible, encouraged verbally to the greatest possible effort. They could terminate the test at any moment if they felt they could no longer maintain the exertion level.

Participants were under cardiopulmonary monitoring during the entire test. Before each CPET, after each change of load and 3 min after the test, 20 µL of blood were collected from the fingertip for determination of lactate concentration (LA) using the Super GL2 analyzer (Müller Gerätebau GmbH, Freital, Germany) calibrated before each series of samples. There were no interruptions in the CPET during the collection of blood samples. During the running test, the triathletes, while running, put their hands on the rail attached to the treadmill and a technician took a blood sample. Before the sample was drawn into the capillary, the first drops of blood were carefully squeezed into a swab. Similarly, during the cycling test, the subject was asked to relax their hands for about 20–30 s before the collection, and then the first drops of blood were discarded before taking the sample into a capillary.

### 2.5. CPET Protocol

The test was terminated by the operator if either VO_2_ or HR showed no further increase with increasing speed/power. The results of the BC analysis and CPET were saved as an Excel (Microsoft Corporation, Washington, United States) spreadsheet for further analysis. The raw data were anonymized and processed with the use of a custom program created in Python software to identify data at AT, RCP and maximum exertion. In accordance with current standards, CPET data were recorded breath by breath and then averaged across 15 s intervals; the highest HR in the interval was recoded, and HR values were not averaged [36]. For statistical evaluation, we included only cases where three of four following criteria were met: RER during test reaching > 1.10, VO_2_ plateau (an increase in VO_2_ with increasing speed/power lower than 100 mL/min), respiratory frequency over 45/min and perceived exertion over 18 in Borg scale [37]. AT and RCP were located from visual inspection. It was assumed that AT was reached after the following criteria were met: (1) VE/VO_2_ curve begins to rise with constant VE/VCO_2_ curve and (2) end-tidal partial pressure of oxygen begins to rise with constant end-tidal partial pressure of carbon dioxide [38]. It was assumed that RCP was reached after the following criteria were met: (1) a decrease in partial pressure of end-tidal CO_2_ (PetCO_2_) after reaching a maximal level; (2) a rapid nonlinear increase in VE (second deflection); (3) the VE/VCO_2_ ratio reached a minimum and began to increase and (4) a nonlinear increase in VCO_2_ versus VO_2_ (departure from linearity) [38].

### 2.6. Retrospective Performance Data

Competition experience was assessed using the enduhub.com (accessed on 15 December 2021) database (Enduhub Corporation, Newark, DE, USA). It is a commonly available website where official scores of participants’ competitions on standardized distances (1/8, 1/4, 1/2 Ironman, Sprinter and Olympic distances of triathlons were included) are uploaded by event organizers. Each score in this database was thoroughly validated by professional companies specialized in time measuring during sports events (Datasport, Szczawno Zdrój, Poland and STS-Timing, Łubianka, Poland). Results are verified before publication by enduhub.com editors. We used the earliest officially available score from a distance-standardized triathlon and used it as a starting point to assess competition experience, which was presented in months (month of competition and CPET were both included). These times were applied to calculate how long each sportsman is engaged in regular training and actively take part in public competitions.

### 2.7. Data Analysis

Statistical analysis has been conducted in R environment/programming language for statistical computing (R Core Team, Vienna, Austria; version 3.6.4;) and lmtest and gtsummary libraries [39,40]. Missing data were identified in lactate values in seven cases and imputation was performed with random forests [41]. Normality was tested with the Anderson-Darling test. Data were calculated as means with SD and 95% confidence intervals (CI). Differences between results of both testing modalities were calculated using paired *t*-tests. A significance level of *p* < 0.05 was adopted for all results.

MLR models were created to evaluate the relationship between the differences in results from treadmill and cycle ergometry (dependent variables), body fat (BF), fat-free mass (FFM) and BMI. Several regression models were initially tested and MLR was selected as the best fit based on the Akaike information criterion. The models were only created for the male population due to group sizes. The Harvey-Collier test was used to test linearity. R-squared (R^2^) was used to assess the quality of the models.

## 3. Results

The differences between CPET results in cycle ergometry and treadmill testing are presented in Table 3 for females and Table 4, Table 5 and Table 6 for males. Results of MLR are presented in Table 7. Only regression results in the two age subgroups are presented, as no significant relationships were found in the whole male population. All statistically significant results are marked bold. Selected (based on highest R^2^) relationships are presented as linear regression graphs (Figure 1); it illustrates the linear relationship between BMI and body fat with CPET parameters in the older population. Moreover, we assessed the effect of theta (θ) interval as well as training advancement and previous experience, and no significant differences were observed for the differences between VO_2_max, VEmax and HRmax. Thus it suggests the homogeneity of our study group.

## 4. Discussion

The hypothesis for the study was mostly confirmed; we demonstrated significant differences in cardiorespiratory parameters at AT, RCP and maximum exertion between cycle ergometry and treadmill testing. Regression models demonstrated significant relationships between BC, BMI and training experience, as well as differences in VO_2_max in cycle ergometry and treadmill testing, especially in the older population of triathletes. However, the coefficient of determination (R^2^) in the regression models ranged from 0.035 to 0.28, which indicates a low regression fit to the observed data. To the best of our knowledge, this is the first study to evaluate the differences between treadmill and cycle ergometer CPET in a large group of triathletes and the first to analyze factors associated with these differences. The results of our study show that both male and female triathletes have a significantly higher VO_2_/AT in the treadmill than in the cycle ergometer tests. The AT is a crucial parameter in determining performance and in training monitoring in endurance sports, as it indicates the level of exertion a triathlete can sustain for a prolonged period of time during competition without rapid lactate build-up [8,42]. Training at the anaerobic threshold (AnT) intensity improves the peak oxygen uptake and the AT level [43]. Recent studies also demonstrate that a large volume of low-intensity training (i.e., below the AT) is important for endurance triathletes [44,45].

Previous research was ambiguous as to whether VO_2AT_ differs between testing modalities in triathletes [8]. The present paper shows significant differences in VO_2AT_. This contradicts the results of several previous studies that found no differences in VO_2_ at AT but were conducted on very small groups (14 participants at most) [8,14,22,25,46]. Some results were similar to our study [21]. The large mean difference in relative VO_2AT_ of 6 mL/kg/min in males shown in our study indicates that the values obtained from both testing modalities likely cannot be used interchangeably. The large difference in HR at AT of 13 bpm in males and 10 bpm in females is a further factor limiting the interchangeability of results from different testing modalities. The large discrepancies would hinder the accurate prescription of low-intensity (below AT) training based on HR zones. This is contrary to results from the studies of Hue and Bolognesi, who found differences of ~7 bpm, but without statistical significance, perhaps due to limited numbers of participants [13,27].

We found that the VO_2/_RCP was significantly higher in treadmills than in cycle ergometry both in males and females, although the differences were smaller than in the AT. We also found a large and significant difference in HR at RCP of 7–10 bpm, again limiting the transferability of results between modalities. To the best of our knowledge, no previous study evaluated RCP in triathletes in treadmill and cycle ergometry testing. The maximum exertion is the most commonly used parameter to assess the aerobic capacity of triathletes [8]. As with the other parameters (VO_2_AT, VO_2_RCP), we found VO_2_max to be significantly higher in treadmill testing than in cycle ergometry. Millet et al. concluded from previous studies that VO_2_max is generally similar in treadmill and cycle ergometry testing in triathletes, and that triathletes’ training adaptation is therefore similar to that of cyclists [8]. This is contrary to the results of our study. Small sample sizes in previous studies are likely the cause for statistically significant differences not having been observed. In the present study, the differences in AT were larger than at maximum exertion. This may explain why significant differences were observed more often at AT than at maximum. The VO_2_max in our studied population is lower than that reported in most previous studies, probably due to the higher mean age of the participants, which is a known factor limiting VO_2_max [8,37]. HRmax was also significantly higher in treadmill testing than in cycle ergometry in both males and females. The results of previous studies on males were conflicting, some indicating HRmax to be lower in cycle ergometry by 6–10 bpm [13,20,21,24] and some finding no significant difference [14,15,18,23,47]. Few studies included females and the evidence was also conflicting; HRmax was observed to be either similar or higher in cycling [25,48].

We found that the VE was higher in cycle ergometry than on the treadmill, despite a slightly lower ventilation frequency, indicating a higher tidal volume. This is contrary to the lower VE and Vf in cyclists at both AT and RCP. The differences may correspond to higher lactate accumulation and an acidosis-induced respiratory response in cycle ergometry at maximum, but not at AT and RCP, where lactate levels were similar. This is partly similar to the findings of Koyal et al., who described a higher respiratory response due to higher acidosis in cycle ergometry during the low, moderate and high intensity of exercise in untrained subjects [49]. The differences at low and moderate intensity are most likely due to our subjects’ experience in triathlon training and therefore lower lactate build-up when cycling at submaximal levels, as it has previously been demonstrated that trained cyclists accumulate less lactate in cycle ergometry [10]. The lower VE and Vf in cycle ergometry at AT and RCP are likely due to the lower VO_2_. Some of the differences between CE and TE may also be due to different breathing patterns in TE and CE. VE increases more steeply in CE, and maximal VE is reached at a lower VT [50]. It has also been shown that cycling leads to a larger decrease in respiratory muscle endurance than using a treadmill, and it has been proposed that the differences in breathing mechanics may be due to a different entrainment of breath in CE and TE [4,30]. Studies showed that triathletes display a higher entrainment of breathing in CE than TE, and that entrainment decreases with increasing load in CE [31]. Altered breath patterns may result in different energy use for breathing and subsequently to differences in lactate levels. Optimizing the breathing patterns could lead to an improved economy in both cycling and running [51]. Overall, the large differences across all measured CPET parameters justify performing two separate tests on CE and TE. The differences may prevent accurate prescription of exercise loads and accurate training progress monitoring in the preparation for competitive events. Further research is needed for swimming.

The variance in differences in VO_2_max obtained from treadmill and cycle ergometer testing was significantly explained by BMI, FATP, FFM and training experience in the group >40 years of age with an R^2^ of 0.25. It has previously been shown that BM and BC are related to aerobic capacity and that the physiological ability to consume oxygen is negatively associated with FM [52,53,54]. However, to the best of our knowledge, this is the first study to demonstrate an association between BC, BMI and the differences in treadmill and cycle ergometry CPET results. We hypothesize that this relationship may be caused by the physiological differences between cycling and running. Cycling is a non-weight-bearing activity, with far less eccentric activity than during running [55]. Therefore, BM and BC may be more important in treadmill testing than in cycle ergometry. It is unclear why the observed differences are better explained by anthropometric variables in the older group compared to the younger group. We may suspect that the difference in the younger population would be better explained by other factors, perhaps by training volume in cycling or running, which were not evaluated in this study. It is also possible that the differences in BC, especially FFM, reflect differences in cardiovascular function, which might play a more important role in older triathletes and could have a different importance in cycle ergometry and treadmill testing [56,57]. Differences in HRmax were significantly explained by BMI and FATP in the older population. It has previously been shown that HRmax is related to BC and BMI, but it remains unclear what the cause for this relationship is or why BC is related to the differences in HRmax between treadmill and cycle ergometry [57].

## 5. Study Limitations

The time intervals between tests varied significantly. Tests were carried out at various times of the season. Training data, except months of experience in triathlon, were unavailable. The female group was too small for meaningful regression analysis, but it was larger than in many previous studies and large enough to observe many significant differences. It was not meaningful to divide females into younger and older age groups, as only 2 females were over 40 years of age. BC was measured with bioelectrical impedance, which may be less accurate than some other methods such as CT/MRI BC analysis, but it is commonly used in sports and therefore may be more practically applicable than more advanced methods. Varied results of CPET may be achieved, and those scores depend on the participant’s endurance level. Preliminary preferred discipline (running or cycling) may also alter our findings. This relates mostly to novice subjects or beginners at lower levels of experience. Future research could address this issue. This study did not evaluate swimming further studies would be needed to determine differences in CPET between TE, CE and swimming.

## 6. Conclusions

VO_2_ and HR are higher in treadmill testing than cycle ergometry at AT, RCP and maximum exertion in both male and female triathletes. The differences are partly explained by BMI, BF and FFM in the population above 40 years of age. These factors were not important in the younger group. The practical implication of this study is that, given the large differences between TE and CE testing, both tests should be carried out in triathletes.

## Figures and Tables

**Figure 1 ijerph-19-03557-f001:**
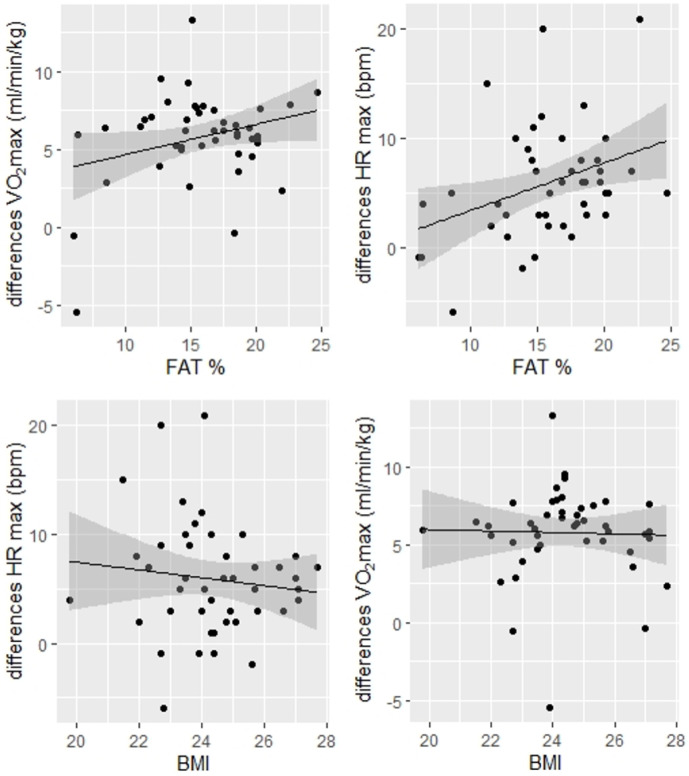
Regression analysis for males in subgroup >40 years. Legend: Multiple linear regression was performed to evaluate the association between differences in treadmill and cycle ergometer test results, and BMI, body fat and fat-free mass in amateur male triathletes. The figure presents the most important (highest R^2^) relationships in the group of males >40 years of age (*n* = 40) as linear regression graphs. Abbreviations: BMI, body mass index (kg/m^2^); HR max, maximal heart rate (bpm); VO_2_ max, maximum oxygen uptake (mL/min/kg).

**Table 1 ijerph-19-03557-t001:** Population characteristics for males and females, including characteristics of the age groups ≤40 and >40 years of age in males and the mean differences between them.

Female Triathletes						
Characteristic	All *n* = 18 ^1^					
Age	33 (7)					
Height	169 (4)					
Weight	61.9 (4.4)					
BMI	21.70 (1.37)					
BF	23.3 (3.5)					
FM	14.55 (2.93)					
FFM	47.38 (2.61)					
Male triathletes						
Characteristic	All *n* = 125	≤40, *n* = 80 ^1^	>40, *n* = 45 ^1^	Difference ^2^	95% CI ^2,3^	*p*-Value ^2^
Age	38 (10)	32 (5)	46 (8)	15.39	12.10–15.67	<0.001
Height	181 (7)	181.0 (7.0)	180.0 (6.0)	1.40	−1.00, 3.80	0.3
Weight	79 (9)	78.0 (10.0)	79.0 (8.0)	−0.67	−4.00, 2.70	0.7
BMI	24.04 (2.19)	23.8 (2.4)	24.4 (1.7)	−0.56	−1.30, 0.18	0.13
BF	15.4 (4.2)	15.2 (4.2)	15.8 (4.3)	−0.60	−2.20, 1.00	0.5
FM	12.3 (4.6)	12.2 (4.8)	12.6 (4.1)	−0.47	−2.10, 1.10	0.6
FFM	66 (6)	66.0 (6.0)	66.0 (6.0)	−0.21	−2.40, 2.00	0.9

^1^ Mean (SD); ^2^ Welch Two Sample *t*-test; ^3^ CI = Confidence Interval. Abbreviations: height (cm); weight (kg); BMI, body masa index; BF, body fat (%); FM, fat mass (kg); FFM, fat-free mass (kg).

**Table 2 ijerph-19-03557-t002:** Participant’s training experience and competition results.

	Males, *n* = 125		Females, *n* = 18	
Distance between CPET	5.29 (6.81)		2.44 (3.10)	
Training experience	103.32 (42.63)		94.11 (38.80)	
**Competition results**				
**Type of competition**	***n* of records**	**Result**	***n* of records**	**Result**
1/8 Iron Man	21	01:18:37 (00:20:38)	1	01:35:27 (00:00:00)
1/4 Iron Man	41	02:29:17 (00:14:31)	6	02:34:18 (00:14:57)
1/2 Iron Man	33	04:57:01 (00:37:04)	4	05:33:17 (00:25:59)
Sprinter’s distance	10	01:09:01 (00:07:13)	2	01:18:28 (00:10:40)
Olympic distance	20	02:33:59 (00:19:57)	5	02:48:43 (00:05:21)

Data are presented as mean (SD). Distance between CPET is presented in days. Training experience is presented in months. All types of competition refer to triathlon distances. The sports result was investigated in a period not longer than three months from the CPET. Competition results are presented as hours:minutes:seconds. Abbreviations: CPET, cardiopulmonary exercise testing.

**Table 3 ijerph-19-03557-t003:** Differences between cycle ergometry (CE) and treadmill (Tr) CPET results in female participants; significant results are bold.

Characteristic	CE, *n* = 18 ^1^	Tr, *n* = 18 ^1^	Difference ^2^	95% CI ^2,3^	*p*-Value ^2^
VO_2AT_	**32.1 (3.8)**	**36.0 (3.6)**	−3.9	−5.4, −2.5	<0.001
VO_2ATa_	**1976 (257)**	**2216 (221)**	−240	−328, −152	<0.001
RER_AT_	0.87 (0.05)	0.88 (0.03)	−0.01	−0.04, 0.02	0.5
HR_AT_	**146 (11)**	**156 (7)**	−10	−14, −6.1	<0.001
O_2_pulse_AT_	**13.64 (2.07)**	**14.28 (1.84)**	−0.65	−1.2, −0.07	0.031
VE_AT_	**53 (7)**	**62 (8)**	−8.5	−12, −4.4	<0.001
RR_AT_	**29 (5)**	**36 (8)**	−6.7	−9.5, −3.9	<0.001
Lac_AT_	1.77 (0.54)	1.87 (0.46)	−0.11	−0.32, 0.10	0.3
VO_2RCP_	**40.2 (4.5)**	**42.9 (4.7)**	−2.8	−4.1, −1.4	<0.001
VO_2RCPa_	**2471 (279)**	**2642 (283)**	−170	−255, −85	<0.001
VCO_2RCP_	**2468 (282)**	**2643 (283)**	−174	−262, −87	<0.001
HR_RCP_	**169 (9)**	**175 (8)**	−6.7	−8.3, −5.1	<0.001
VE_RCP_	80 (10)	86 (9)	−5.3	−11, 0.32	0.063
RR_RCP_	**38 (6)**	**44 (7)**	−5.7	−7.7, −3.6	<0.001
O_2_pulse_RCP_	**14.69 (1.80)**	**15.13 (1.98)**	−0.44	−0.82, −0.05	0.029
Lac_RCP_	4.30 (0.69)	4.50 (0.82)	−0.20	−0.72, 0.31	0.4
VO_2_max	**43.9 (5.2)**	**45.3 (4.9)**	−1.4	−2.8, −0.02	0.047
VO_2_max_a_	**2704 (319)**	**2791 (305)**	−87	−169, −4.2	0.040
RERmax	1.13 (0.03)	1.11 (0.03)	0.01	−0.01, 0.04	0.3
O_2_pulse_max_	15.12 (1.96)	15.33 (2.03)	−0.20	−0.59, 0.18	0.3
HRmax	**179 (10)**	**183 (9)**	−3.5	−5.2, −1.8	<0.001
VEmax	110 (18)	104 (13)	6.0	−1.8, 14	0.12
RRmax	55 (11)	55 (9)	−0.11	−2.4, 2.2	>0.9
Lacmax	**10.76 (1.94)**	**9.40 (1.28)**	1.4	0.47, 2.2	0.005

^1^ Mean (SD); ^2^ Paired *t*-test; ^3^ CI = Confidence Interval. Abbreviations: VO_2,_ oxygen uptake; AT, anaerobic threshold; VO_2AT_, relative VO_2_ at AT (mL/kg/min); VO_2ATa_, absolute VO_2_ at AT (mL/min); RER_AT_, respiratory exchange ratio at AT; HR_AT_, heart rate at AT (bpm); O_2_pulse_AT_, oxygen pulse at AT (mL/beat); VE_AT_, pulmonary ventilation at AT (L/min); RR_AT_, respiratory rate at AT (breaths per minute); Lac_AT_, lactate concentration at AT (mmol/L); RCP, respiratory compensation point; VO_2RCP_, relative VO_2_ at RCP (mL/kg/min); VO_2RCPa_, absolute VO_2_ at RCP (mL/min); VCO_2RCP_, carbon dioxide production at RCP(mL/min); HR_RCP_, heart rate at RCP (bpm); VE_RCP_, pulmonary ventilation at RCP(L/min); RR_RCP_, respiratory rate at RCP (breaths per minute); O_2_pulse_RCP_, oxygen pulse at RCP (mL/beat); Lac_RCP_, lactate concentration at RCP(mmol/L); VO_2_max, relative maximum VO_2_ (mL/kg/min); VO_2_max_a_, absolute maximum VO_2_ (mL/min); RERmax, maximal respiratory exchange ratio; O_2_pulsemax, maximal oygen pulse (mL/beat); HRmax, maximal heart rate (bpm); VEmax, maximal pulmonary ventilation (L/min); RRmax, maximal respiratory rate (breaths per minute); Lacmax, maximal lactate concentration (mmol/L).

**Table 4 ijerph-19-03557-t004:** Differences between cycle ergometry (CE) and treadmill (TE) CPET results in male participants; significant results are bold.

Characteristic	CE, *n* = 125 ^1^	TE, *n* = 125 ^1^	Difference ^2^	95% CI ^2,3^	*p*-Value ^2^
VO_2AT_	**32.8 (5.4)**	**38.8 (4.6)**	−6.0	−6.7, −5.3	<0.001
VO_2ATa_	**2530 (366)**	**3012 (345)**	−482	−541, −424	<0.001
RER_AT_	**0.86 (0.06)**	**0.89 (0.04)**	−0.02	−0.04, −0.01	<0.001
HR_AT_	**136 (11)**	**149 (10)**	−13	−15, −11	<0.001
O_2_pulse_AT_	**18.73 (2.88)**	**20.32 (2.36)**	−1.6	−1.9, −1.2	<0.001
VE_AT_	**66 (10)**	**83 (11)**	−17	−18, −15	<0.001
RR_AT_	**28 (5)**	**37 (8)**	−8.2	−9.3, −7.1	<0.001
Lac_AT_	**1.70 (0.37)**	**1.77 (0.40)**	−0.07	−0.14, 0.00	0.043
VO_2RCP_	**44 (8)**	**48 (6)**	−4.1	−5.1, −3.1	<0.001
VO_2RCPa_	**3372 (533)**	**3707 (420)**	−335	−411, −259	<0.001
VCO_2RCP_	**3378 (539)**	**3708 (420)**	−329	−406, −252	<0.001
HR_RCP_	**163 (10)**	**172 (9)**	−9.4	−11, −8.2	<0.001
VE_RCP_	**110 (17)**	**119 (15)**	−9.4	−12, −6.9	<0.001
RR_RCP_	**39 (7)**	**46 (10)**	−6.7	−8.1, −5.2	<0.001
O_2_pulse_RCP_	**20.78 (3.44)**	**21.59 (2.65)**	−0.81	−1.2, −0.39	<0.001
Lac_RCP_	4.21 (0.60)	4.29 (0.74)	−0.08	−0.23, 0.06	0.3
VO_2_max	**49 (7)**	**52 (6)**	−2.9	−3.6, −2.1	<0.001
VO_2_max_a_	**3808 (476)**	**4045 (435)**	−238	−300, −175	<0.001
RERmax	**1.13 (0.04)**	**1.11 (0.03)**	0.02	0.02, 0.03	<0.001
O_2_pulse_max_	**21.54 (2.91)**	**22.16 (2.66)**	−0.62	−0.94, −0.29	<0.001
HRmax	**177 (10)**	**183 (10)**	−5.9	−6.8, −5.0	<0.001
VEmax	**158 (24)**	**152 (18)**	6.6	3.4, 9.8	<0.001
RRmax	**57 (10)**	**58 (10)**	−1.8	−3.6, −0.05	0.044
Lacmax	**10.93 (1.82)**	**9.68 (1.65)**	1.2	0.91, 1.6	<0.001

^1^ Mean (SD); ^2^ Welch Two Sample *t*-test; ^3^ CI = Confidence Interval. Abbreviations: VO_2,_ oxygen uptake; AT, anaerobic threshold; VO_2AT_, relative VO_2_ at AT (ml/kg/min); VO_2ATa_, absolute VO_2_ at AT (ml/min); RER_AT_, respiratory exchange ratio at AT; HR_AT_, heart rate at AT (bpm); O_2_pulse_AT_, oxygen pulse at AT (ml/beat); VE_AT_, pulmonary ventilation at AT (L/min); RR_AT_, respiratory rate at AT (breaths per minute); Lac_AT_, lactate concentration at AT (mmol/L); RCP, respiratory compensation point; VO_2RCP_, relative VO_2_ at RCP (ml/kg/min); VO_2RCPa_, absolute VO_2_ at RCP (ml/min); VCO_2RCP_, carbon dioxide production at RCP(ml/min); HR_RCP_, heart rate at RCP (bpm); VE_RCP_, pulmonary ventilation at RCP(L/min); RR_RCP_, respiratory rate at RCP (breaths per minute); O_2_pulse_RCP_, oxygen pulse at RCP (ml/beat); Lac_RCP_, lactate concentration at RCP(mmol/L); VO_2_max, relative maximum VO_2_ (ml/kg/min); VO_2_max_a_, absolute maximum VO_2_ (ml/min); RERmax, maximal respiratory exchange ratio; O_2_pulsemax, maximal oygen pulse (ml/beat); HRmax, maximal heart rate (bpm); VEmax, maximal pulmonary ventilation (L/min); RRmax, maximal respiratory rate (breaths per minute); Lacmax, maximal lactate concentration (mmol/L).

**Table 5 ijerph-19-03557-t005:** Differences between cycle ergometry (CE) and treadmill (TE) CPET results in male participants ≤40 years; significant results are bold.

Characteristic	CE, *n* = 80 ^1^	TE, *n* = 80 ^1^	Difference ^2^	95% CI ^2,3^	*p*-Value ^2^
VO_2AT_	**34 (6)**	**40 (5)**	−6.2	−7.2, −5.1	<0.001
VO_2ATa_	**2584 (363)**	**3078 (350)**	−494	−577, −412	<0.001
RER_AT_	**0.87 (0.06)**	**0.88 (0.04)**	−0.02	−0.03, 0.00	0.047
HR_AT_	**140 (11)**	**151 (10)**	−12	−14, −9.5	<0.001
O_2_pulse_AT_	**18.60 (2.86)**	**20.40 (2.41)**	−1.8	−2.3, −1.3	<0.001
VE_AT_	**66 (10)**	**82 (11)**	−17	−19, −14	<0.001
RR_AT_	**28 (6)**	**36 (7)**	−8.2	−9.5, −6.9	<0.001
Lac_AT_	1.67 (0.37)	1.76 (0.40)	−0.08	−0.17, 0.00	0.057
VO_2RCP_	**45 (7)**	**49 (6)**	−4.0	−5.0, −2.9	<0.001
VO_2RCPa_	**3461 (447)**	**3790 (417)**	−328	−415, −242	<0.001
VCO_2RCP_	**3463 (458)**	**3791 (417)**	−328	−417, −239	<0.001
HR_RCP_	**166 (10)**	**175 (9)**	−8.9	−10, −7.4	<0.001
VE_RCP_	**109 (16)**	**119 (15)**	−10	−13, −7.0	<0.001
RR_RCP_	**38 (7)**	**45 (9)**	−6.4	−8.0, −4.9	<0.001
O_2_pulse_RCP_	20.90 (2.94)	21.71 (2.69)	−0.81	−1.3, −0.36	<0.001
Lac_RCP_	4.15 (0.63)	4.24 (0.72)	−0.08	−0.26, 0.10	0.4
VO_2_max	**51 (7)**	**53 (7)**	−2.8	−3.8, −1.7	<0.001
VO_2_max_a_	**3885 (471)**	**4118 (418)**	−233	−316, −149	<0.001
RERmax	**1.13 (0.04)**	**1.11 (0.03)**	0.02	0.01, 0.03	<0.001
O_2_pulse_max_	21.65 (2.94)	22.22 (2.62)	−0.57	−1.0, −0.14	0.010
HRmax	**180 (9)**	**186 (9)**	−5.9	−7.0, −4.8	<0.001
VEmax	**159 (24)**	**152 (16)**	7.3	2.9, 12	0.001
RRmax	56 (11)	58 (8)	−1.3	−3.3, 0.69	0.2
Lacmax	**10.94 (1.91)**	**9.81 (1.53)**	1.1	0.59, 1.7	<0.001

^1^ Mean (SD); ^2^ Welch Two Sample *t*-test; ^3^ CI = Confidence Interval. Abbreviations: VO_2,_ oxygen uptake; AT, anaerobic threshold; VO_2AT_, relative VO_2_ at AT (ml/kg/min); VO_2ATa_, absolute VO_2_ at AT (ml/min); RER_AT_, respiratory exchange ratio at AT; HR_AT_, heart rate at AT (bpm); O_2_pulse_AT_, oxygen pulse at AT (ml/beat); VE_AT_, pulmonary ventilation at AT (L/min); RR_AT_, respiratory rate at AT (breaths per minute); Lac_AT_, lactate concentration at AT (mmol/L); RCP, respiratory compensation point; VO_2RCP_, relative VO_2_ at RCP (ml/kg/min); VO_2RCPa_, absolute VO_2_ at RCP (ml/min); VCO_2RCP_, carbon dioxide production at RCP(ml/min); HR_RCP_, heart rate at RCP (bpm); VE_RCP_, pulmonary ventilation at RCP(L/min); RR_RCP_, respiratory rate at RCP (breaths per minute); O_2_pulse_RCP_, oxygen pulse at RCP (ml/beat); Lac_RCP_, lactate concentration at RCP(mmol/L); VO_2_max, relative maximum VO_2_ (ml/kg/min); VO_2_max_a_, absolute maximum VO_2_ (ml/min); RERmax, maximal respiratory exchange ratio; O_2_pulsemax, maximal oygen pulse (ml/beat); HRmax, maximal heart rate (bpm); VEmax, maximal pulmonary ventilation (L/min); RRmax, maximal respiratory rate (breaths per minute); Lacmax, maximal lactate concentration (mmol/L).

**Table 6 ijerph-19-03557-t006:** Differences between cycle ergometry (CE) and treadmill (TE) CPET results in male participants > 40 years; significant results are bold.

Characteristic	CE, *n* = 45 ^1^	TE, *n* = 45 ^1^	Difference ^2^	95% CI ^2,3^	*p*-Value ^2^
VO_2AT_	**31.1 (4.6)**	**36.9 (3.1)**	−5.8	−6.6, −4.9	<0.001
VO_2ATa_	**2434 (354)**	**2895 (305)**	−461	−532, −390	<0.001
RER_AT_	**0.86 (0.05)**	**0.89 (0.04)**	−0.03	−0.05, −0.02	<0.001
HR_AT_	**129 (10)**	**144 (9)**	−15	−17, −12	<0.001
O_2_pulse_AT_	**18.96 (2.93)**	**20.18 (2.29)**	−1.2	−1.7, −0.75	<0.001
VE_AT_	**67 (11)**	**83 (10)**	−16	−19, −14	<0.001
RR_AT_	**29.1 (4.7)**	**37.3 (8.4)**	−8.2	−10, −6.2	<0.001
Lac_AT_	1.75 (0.36)	1.80 (0.39)	−0.05	−0.16, 0.07	0.4
VO_2RCP_	**41.1 (8.4)**	**45.4 (4.3)**	−4.3	−6.2, −2.4	<0.001
VO_2RCPa_	**3213 (634)**	**3559 (387)**	−346	−494, −198	<0.001
VCO_2RCP_	**3228 (637)**	**3560 (387)**	−332	−481, −182	<0.001
HR_RCP_	**157 (9)**	**167 (8)**	−10	−12, −8.3	<0.001
VE_RCP_	**112 (19)**	**119 (16)**	−7.8	−12, −3.7	<0.001
RR_RCP_	**40 (7)**	**47 (11)**	−7.1	−10, −4.1	<0.001
O_2_pulse_RCP_	20.55 (4.21)	21.36 (2.59)	−0.82	−1.7, 0.08	0.073
Lac_RCP_	4.31 (0.53)	4.40 (0.78)	−0.09	−0.36, 0.19	0.5
VO_2_max	**46.9 (6.0)**	**49.9 (4.8)**	−3.0	−4.2, −1.8	<0.001
VO_2_max_a_	**3670 (459)**	**3916 (439)**	−246	−343, −149	<0.001
RERmax	**1.13 (0.03)**	**1.11 (0.03)**	0.03	0.01, 0.04	<0.001
O_2_pulse_max_	21.35 (2.88)	22.05 (2.76)	−0.71	−1.2, −0.20	0.008
HRmax	**172 (8)**	**178 (9)**	−5.9	−7.5, −4.3	<0.001
VEmax	157 (25)	151 (22)	5.5	1.0, 9.9	0.018
RRmax	57 (10)	59 (12)	−2.7	−6.6, −4.9	<0.001
Lacmax	**10.90 (1.65)**	**9.45 (1.82)**	1.5	0.72, 2.2	<0.001

^1^ Mean (SD); ^2^ Welch Two Sample *t*-test; ^3^ CI = Confidence Interval. Abbreviations: VO_2,_ oxygen uptake; AT, anaerobic threshold; VO_2AT_, relative VO_2_ at AT (ml/kg/min); VO_2ATa_, absolute VO_2_ at AT (ml/min); RER_AT_, respiratory exchange ratio at AT; HR_AT_, heart rate at AT (bpm); O_2_pulse_AT_, oxygen pulse at AT (ml/beat); VE_AT_, pulmonary ventilation at AT (L/min); RR_AT_, respiratory rate at AT (breaths per minute); Lac_AT_, lactate concentration at AT (mmol/L); RCP, respiratory compensation point; VO_2RCP_, relative VO_2_ at RCP (ml/kg/min); VO_2RCPa_, absolute VO_2_ at RCP (ml/min); VCO_2RCP_, carbon dioxide production at RCP(ml/min); HR_RCP_, heart rate at RCP (bpm); VE_RCP_, pulmonary ventilation at RCP(L/min); RR_RCP_, respiratory rate at RCP (breaths per minute); O_2_pulse_RCP_, oxygen pulse at RCP (ml/beat); Lac_RCP_, lactate concentration at RCP(mmol/L); VO_2_max, relative maximum VO_2_ (ml/kg/min); VO_2_max_a_, absolute maximum VO_2_ (ml/min); RERmax, maximal respiratory exchange ratio; O_2_pulsemax, maximal oygen pulse (ml/beat); HRmax, maximal heart rate (bpm); VEmax, maximal pulmonary ventilation (L/min); RRmax, maximal respiratory rate (breaths per minute); Lacmax, maximal lactate concentration (mmol/L).

**Table 7 ijerph-19-03557-t007:** Regression analysis for males in two subgroups (≤40 years and >40 years).

Age Group	≤40 Years			>40 Years		
Predictors	BMI	BF	FFM	BMI	BF	FFM
VO_2_AT						
b	0.02	0.15	0.01	−0.92	0.39	0.16
95% CI	−0.94, 1.0	−0.33, 0.63	−0.19, 0.22	−1.7, −0.15	0.13, 0.64	−0.03, 0.34
*p*-value	>0.9	0.5	>0.9	0.020	0.004	0.089
R^2^	0.023			0.197		
HRAT						
b	0.23	−0.10	−0.06	−1.3	1.0	0.17
95% CI	−1.8, 2.3	−1.1, 0.93	−0.49, 0.38	−3.5, 0.93	0.29, 1.8	−0.36, 0.71
*p*-value	0.8	0.9	0.8	0.2	0.008	0.5
R^2^	0.001			0.172		
VO_2_RCP						
b	−0.16	0.30	0.05	−1.4	0.80	0.15
95% CI	−1.1, 0.82	−0.19, 0.80	−0.16, 0.26	−3.1, 0.26	0.24, 1.4	−0.25, 0.56
*p*-value	0.8	0.2	0.6	0.095	0.006	0.4
R^2^	0.055			0.169		
HRRCP						
b	0.47	−0.16	−0.01	−1.3	0.79	0.13
95% CI	−1.0, 1.9	−0.89, 0.57	−0.32, 0.30	−3.0, 0.34	0.24, 1.4	−0.27, 0.54
*p*-value	0.5	0.7	>0.9	0.11	0.006	0.5
R^2^	0.008			0.035		
VO_2_max						
b	0.35	0.11	0.09	1.3	0.47	0.24
95% CI	−0.60, 1.3	−0.37, 0.59	−0.29, 0.12	−2.4, −0.19	0.11, 0.83	−0.03, 0.50
*p*-value	0.5	0.6	0.4	0.022	0.012	0.078
R^2^	0.054			0.163		
HRmax						
b	−0.13	0.10	0.12	−1.5	0.78	−0.01
95% CI	−1.1, 0.87	−0.40, 0.61	−0.09, 0.31	−2.8, −0.16	0.35, 1.2	−0.32, 0.30
*p*-value	0.8	0.7	0.3	0.028	<0.001	0.9
R^2^	0.028			0.28		

Abbreviations: BMI, body masa index; BF, body fat (%); FFM, fat-free mass (kg); VO_2_AT, oxygen consumption at anaerobic threshold (ml/kg/min); HRAT, heart rate at anaerobic threshold (bpm); VO2RCP, oxygen consumption at respiratory compensation point(ml/kg/min); HRRCP, heart rate at respiratory compensation point(bpm); VO_2_max, maximum oxygen uptake (ml/kg/min); HRmax, maximal heart rate (bpm).

## Data Availability

The datasets generated during and/or analyzed during the current study are available from the corresponding author on reasonable request.

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
