# Peer review of "Differences between Treadmill and Cycle Ergometer Cardiopulmonary Exercise Testing Results in Triathletes and Their Association with Body Composition and Body Mass Index"

_ijerph, 2022, doi:10.3390/ijerph19063557_

Round 1

Reviewer 1 Report

I appreciate the efforts of the authors in creating the article. The topic is very interesting and the number of participants is above standard. After the revision, the article has been improved, after incorporating comments, the article is very inspiring and more informative. However, there are still facts that need to be corrected and supplemented in order to remove some question marks.

Major issues:

For a better understanding of the chapter “Materials and methods”, I recommend that this section be divided into other sub-chapters including “Subjects” and for example “Study design” or “Measures”.

If the order of the tests was not randomized, that fact should be included in the limits of the study.

Line 110-114: The aim of the work needs to be better specified. What athletes (triathletes) and what level? Are all performance levels evenly covered, including both beginners and Olympic levels, so that it can be argued that this is generally true for triathletes? I believe that at different performance levels the results can be different. Table 6 shows the athletes' performance, but unfortunately the table is in PDF including the marked modifications, and it is not easy to understand. It is not clear whether the performance level of all athletes or only the best ones is given. I believe that this table should rather be in the section "Material and methods" as a characteristic of the participant population. Once completed, this could provide the required overview of the participants in the study. What matters is what performance levels of triathletes the research covers.

Line: 402-403: In connection with the need to specify the aim of the work, the conclusion should also be adjusted. The findings of the study should be related to the given level of triathletes, as different results of CPET tests on individual ergometers can be assumed at different levels. Apart from the fact that there are different types of triathletes (according to the preferred discipline). Especially for beginers, this may play a role. I also recommend mentioning this in the paper.

Line 166: I recommend referring to the selected warm up design. In my opinion this exercise may not be sufficient, there are other variants, with a break after a slightly more intense exercise. But it is comparable for both tests, if quoted, it is acceptable.

Minor comments:

Line 3339: VO2/RCP – slash is in the upper index

Author Response

Dear Editor,   

Dear Reviewers, 

Thank you very much for the thorough analysis of our manuscript, for your valuable and helpful comments and for giving us the opportunity to revise and improve our submission. We hope that our replies and explanations, as well as the amendments to the manuscript, fully address your concerns. We keep the change tracking. In the following, please find our answers to your comments.

I appreciate the efforts of the authors in creating the article. The topic is very interesting and the number of participants is above standard. After the revision, the article has been improved, after incorporating comments, the article is very inspiring and more informative. However, there are still facts that need to be corrected and supplemented in order to remove some question marks.

Major issues:

For a better understanding of the chapter “Materials and methods”, I recommend that this section be divided into other sub-chapters including “Subjects” and for example “Study design” or “Measures”.

Response: Thank you for your valuable attention. We improved the structure of our manuscript.

If the order of the tests was not randomized, that fact should be included in the limits of the study.

Response: It was randomly chosen by the athlete. 

Line 110-114: The aim of the work needs to be better specified. What athletes (triathletes) and what level? Are all performance levels evenly covered, including both beginners and Olympic levels, so that it can be argued that this is generally true for triathletes? I believe that at different performance levels the results can be different. Table 6 shows the athletes' performance, but unfortunately the table is in PDF including the marked modifications, and it is not easy to understand. It is not clear whether the performance level of all athletes or only the best ones is given. I believe that this table should rather be in the section "Material and methods" as a characteristic of the participant population. Once completed, this could provide the required overview of the participants in the study. What matters is what performance levels of triathletes the research covers.

Response: The athletes were of all levels, but due to the retrospective nature of the paper, we were not able to evenly select groups with various experience. Differences in specific groups would be an excellent topic for further prospective studies. The level of the participants can be evaluated by training experience, race times and CPET results, it may be clearly seen that a wide variety of triathletes were taken into account. We verified the effect of training experience and the interval between the tests, and no significant differences were found in cardiovascular and respiratory parameters such as VO2max, VEmax or Hrmax. We are not able to provide more data than this and we believe that the background we provided is relatively in-depth for a retrospective study.

Line: 402-403: In connection with the need to specify the aim of the work, the conclusion should also be adjusted. The findings of the study should be related to the given level of triathletes, as different results of CPET tests on individual ergometers can be assumed at different levels. Apart from the fact that there are different types of triathletes (according to the preferred discipline). Especially for beginers, this may play a role. I also recommend mentioning this in the paper.

Response: We agree that these are limitations of the study and we added this information to the limitations section. Prospective cohort studies would be needed to allow such precise analyses.

Line 166: I recommend referring to the selected warm up design. In my opinion this exercise may not be sufficient, there are other variants, with a break after a slightly more intense exercise. But it is comparable for both tests, if quoted, it is acceptable.

Response: The warmup was comparable for both tests and was designed so as to impact the results as little as possible. This does not impact the described differences between tests. We proposed to change the term "warm-up" to "adaptation" 

Minor comments:

Line 3339: VO2/RCP – slash is in the upper index

We would like to thank  the  Reviewer for his systematic and thorough analysis of our paper, and for his further suggestions. We believe that all remaining concerns are now fully addressed.

Reviewer 2 Report

Dear authors,

The paper is very interesting, and it has been improved since the last submission. I still feel that some revision should be made before considering it for publication. Please check my comments:

The authors mentioned that “We agree that in certain cases it could account for a significant difference. It must however be noted that the median interval between tests was 1 day, the mean approx.. 5 days, and only a few cases had an interval of over a month. We decided that it would be more valuable to include these cases and increase the sample size rather than to omit them for a more homogenous sample.”

I suggest that this information should be added. Did you check if the cases with an interval over than a month influence the results? Nonetheless, please include how many cases were included with an interval over than a month).

Abstract

L26 - please add minimum and maximum ages for female.

Keywords - Please avoid repeating the same words that are present in the title.

Introduction

L55 – AT - please check the letter size.

L62-63 - Please check the sentence to avoid repetition of "used" word.

L66-68 - This statement is correct, but it does not support their use in athletes. I suggest adjusting in order to be more suited for athletes.

L70-71 - Again, the same comment. Your work is about amateur athletes. Thus, I would suggest keeping the rationale for such individuals

L113 - Please add the hypothesis of the study.

Table 1 - What was the rationale to divide male population </>40 years old? This should be also reflected in the introduction section

L240-242 - After finding this explanation, I would suggest that this information could be used before when this division is introduced. Also, introduction section should also approach this explanation. Please try to use more updated references to support this division

Results

L250-251- please add some sentence to interpret figure 1.

Discussion

I suggest changing the beginning of your discussion.

It should start with the aim of your study and the main results.

L328-338 - The paragraph presents several ideas, but it too long and confusing. Please split the ideas that provide a better organization to share them. In the current form, it is difficult for the reader

L348-349 - check the English grammar. It should be improved.

L358 – “Few studies included 357 females and the evidence was also conflicting 25,45”. Why? Please add more details as you did in the previous sentences

L400 - I suggest adding a practical implication from this study

Conclusion

L404-405 - This idea can be reinforced in discussion. Please consider this.

L406-407 - What about the others? I know the results did not find explanations for others, but this should be stated in conclusion

Study limitations

I suggest presenting this part before conclusion section. please add suggestions for future research.

General comment - There several paragraphs too long that needs revision. Some of them are hard to read.

Best regards

Author Response

Dear Editor,   

Dear Reviewers, 

Thank you very much for the thorough analysis of our manuscript, for your valuable and helpful comments and for giving us the opportunity to revise and improve our submission. We hope that our replies and explanations, as well as the amendments to the manuscript, fully address your concerns. We keep the change tracking. In the following, please find our answers to your comments.

Dear authors,

The paper is very interesting, and it has been improved since the last submission. I still feel that some revision should be made before considering it for publication. Please check my comments:

The authors mentioned that “We agree that in certain cases it could account for a significant difference. It must however be noted that the median interval between tests was 1 day, the mean pprox... 5 days, and only a few cases had an interval of over a month. We decided that it would be more valuable to include these cases and increase the sample size rather than to omit them for a more homogenous sample.”

I suggest that this information should be added. Did you check if the cases with an interval over than a month influence the results? Nonetheless, please include how many cases were included with an interval over than a month).

Response: 23 cases had an interval of over 1 month. We performed an additional analysis that demonstrated that these intervals had no significant effect on the observed differences.

Abstract

L26 - please add minimum and maximum ages for female.

Response:Done

Keywords - Please avoid repeating the same words that are present in the title.

Response: We add new keywords

Introduction

L55 – AT - please check the letter size.

Response:Done

L62-63 - Please check the sentence to avoid repetition of "used" word.

Response:Done

L66-68 - This statement is correct, but it does not support their use in athletes. I suggest adjusting in order to be more suited for athletes.

Response:Indeed, athletes should be accustomed to treadmill training. This fragment was omitted. ECG remains an important factor for athletes.

L70-71 - Again, the same comment. Your work is about amateur athletes. Thus, I would suggest keeping the rationale for such individuals

Response:Done

L113 - Please add the hypothesis of the study.

Response:Done

Table 1 - What was the rationale to divide male population </>40 years old? This should be also reflected in the introduction section

L240-242 - After finding this explanation, I would suggest that this information could be used before when this division is introduced. Also, introduction section should also approach this explanation. Please try to use more updated references to support this division

Response: Thank you for your valuable and significant comment here. According to your suggestion, we revised our introduction paragraph. 

Results

L250-251- please add some sentence to interpret figure 1.

Response:Done

Discussion

I suggest changing the beginning of your discussion.

It should start with the aim of your study and the main results.

Response:Done

L328-338 - The paragraph presents several ideas, but it too long and confusing. Please split the ideas that provide a better organization to share them. In the current form, it is difficult for the reader

Response:The paragraph has been divided and we improved clarity.

L348-349 - check the English grammar. It should be improved.

Response:Done

L358 – “Few studies included 357 females and the evidence was also conflicting 25,45”. Why? Please add more details as you did in the previous sentences

Response:Done

L400 - I suggest adding a practical implication from this study

Response:One of the conclusions is a practical implication, we highlighted this for clarity.

Conclusion

L404-405 - This idea can be reinforced in discussion. Please consider this.

Response:We added a paragraph in the discussion. 

L406-407 - What about the others? I know the results did not find explanations for others, but this should be stated in conclusion

Response:Done

Study limitations

I suggest presenting this part before conclusion section. please add suggestions for future research.

Response:Done

General comment - There several paragraphs too long that needs revision. Some of them are hard to read.

Several paragraphs have been revised.

Once more,  we would like to thank the Reviewer for important comments and for input on our report. We believe that all remaining concerns are now fully addressed.

Round 2

Reviewer 1 Report

Dear authors, requested comments have been incorporated, questions have been answered.

Minor comments:
Apart from minor formal errors in the text, the number of competitors in table 2 does not match, the sum of individual competitors does not correspond to the total number of 125 (125 ≠ 21+42+33+10+20) for men, nor 18 (18 ≠ 1+5+4+2+5) for women. If the error is not only in one of the sub-numbers, but in the total, it must be corrected in the other tables and in the text.

Author Response

The suggested change is made directly in the manuscript. 

This manuscript is a resubmission of an earlier submission. The following is a list of the peer review reports and author responses from that submission.

Round 1

Reviewer 1 Report

The main aim of the paper: „Differences between treadmill and cycle ergometer cardiopul-monary exercise testing results in amateur triathletes and their association with body composition and body mass index.“ was to assess the difference in VO2 and HR at maximum and at submaximal exertion in cycle ergometry and treadmill testing. A further aim was to evaluate whether an association exists between these parameters and body composition or body mass index.

I would like to appretiate the efforts of the authors, the topic is interesting, but there are some esential points, that cannot be used in this way, it is needed improve or change them.

Major issues:

The crucial problem is the interval between the bicycle test and the treadmill test. It is not possible to use two tests at different times in the training cycle to assess the difference between the tests, the optimum for all participants is the same, reasonably short, spacing between the tests for all participants, to ensure that all participants in both tests are at the same training level. Randomization of the test order is also essential. Some of these comments are made within the limits of the work, but I believe this is not enough, the study thus constructed does not provide objective and valid information.

Particular issues:

Line 115 Triathlon is a very complex discipline due to the presence of 3 sports. The three-month period is not long enough for an individual to qualify as a triathlete and the results are more likely to be linked to their previous specialization (running, cycling, and swimming). There are studies comparing the VO2 protocol on the treadmill and bicycle of the last 3 years which demonstrate significant differences between runners, cyclists and swimmers.

Line 116 The two-month period between tests can be decisive for results if an individual were to train for 3 months, take the first test, and after two months the second, the training time is nearly double. Another problem for more experienced athletes would be if one of the tests was in transition.

Line 117 Do you have any information about the level of competitors tested and the individual level of their individual disciplines? The discussion noted that baseline data on training volume in each discipline was not assessed. But these informations may be related to the results. This would certainly be useful for assessing results.

Line 127 Questions: In what proportion was the first test taken? 50:50 treadmill/bicycle? Was the order of tests randomized?

Line 152 Walking or pedaling with no resistence as a warm-up for experience athlete may be not enought

Line 181 Indeed, all tests have achieved a plateau. A RER of more than 1.10 is standard as a condition, but its value is very individual.

Line 198 Missing )

Line 236 The VT parameter was not evaluated? The results show that it scores higher on the bicycle than on the treadmill.

Line 322 Many previous studies have shown that the breathing pattern differs between exercise on the bicycle and treadmill. The position of the body on the work of the respiratory muscles has a major impact. Again, search for latest comparison of a Treadmill Vs. Cycling Protocol.

Line 340 Again, the body position when cycling and running speed has a major impact on tidal volume and breathing frequency. It was proven in previous studies.

Line 364 This is a fundamental problem because if you check previous studies you can find that cyclists have a significantly higher VO2peak during cycling.

Line 372 The conclusion should be better worded.

Reviewer 2 Report

This is a very interesting study. The design and significance of this study are great. I have some suggestions below. 

  • Line 24: I assume the authors meant to say the differences “between Re and CE” in HR and VO2max? If so, please clarify in the sentence.  
  • Line 199: Please define PetCo2. 
  • Line 295: Although the P values are below 0.05, the R-square values are not high enough to call them correlated. Please consider rephrasing this sentence to eliminate misinterpretation.  
  • It may be the journal's technical issue, but the resolution (picture quality) of Figure 1 needs to be higher.  

Reviewer 3 Report

Dear Editor,

The paper is very interesting however I only read it until the methods section because I found major issues. Specifically, the following paragraph states:

L111-117 – “The data of participants was gathered from records of commercial CPET performed in years 2013–2020. They were recruited via internet and social media advertisements, or via recommendation from trainers or other clients. The tests were carried out on personal request of the participants as part of training optimization and diagnostics. The participants were amateur triathletes who had participated in competitive events. Inclusion criteria for the study were: age over 18 years, triathlon training for at least 3 months, having a treadmill test and a cycle ergometer test performed within a maximum 2 months’ timeframe, and meeting the maximum exertion criteria described below.“

These seems to be the major issues of this work.

The fact that the data was collected between 2013 and 2020 is not a problem, but was the same CPET used? Were the same procedures to access VO2 max applied? What was the strength and conditioning of the participants at the moments of the assessment?

In addition, the fact that 2 months were given to perform the 2 assessment is a major concern. Since the participants of this study are amateur athletes, they can probably perform 1-3 training sessions a week, which provides different conditions for all. In addition, some participants maybe have improved they capacity from the 1st assessment until the 2nd assessment which will negatively influence the results. Even if these issues are reported in the limitation section, this is a major concern. I would suggest to authors reconsider the entire methodology and only include participants that performed the 2 tests within 1-2 weeks. Otherwise, all results could have a training interference.

A sentence such “Identical study methods and procedures were used during the entire period from which data were gathered” is not enough to explain all this issues.

I must add the introduction was well developed and the rationale of the study was well explained. The only suggestion is to add a paragraph to highlight the importance of body composition for triathletes and their relationship with VO2 which was not developed enough.

For all the reasons presented, I have to reject the paper for publication and ask the authors to reconsider their submission after major revisions.

Thank you